# Expedient access to bora-butenolide bioisosteres by counteranion-mediated *trans*-hydroboration of alkynes

Yuan-Wen Liu[1,4], Yu Liu[2,4], Yanting Zheng[1], Mengfan Zhang[1], Meng-En Ren[1], Peiyu Hua[1], Jie Han [2] ✉, Alois Fürstner [3] ✉ & Hongming Jin [1] ✉

The hydroboration of alkynes is a textbook example of a *syn*-selective concerted addition reaction, while *trans*-selective additions of borane to alkynes remain to be developed. We herein report a transition metal-free *anti*-addition of pinacolborane to alkynes, facilitated by the counteranion effect. This work further develops Chan alkyne reduction by utilizing the borane instead of aluminohydride reagents, enabling the facile synthesis of valuable five-membered boracycles that constitute isosteric alternatives to bioactive butenolides and a versatile platform for abundant downstream transformations. The practical method is distinguished by excellent regioselectivity, a broad substrate scope, and high compatibility with a variety of functional groups. The exploration of *trans*-selective patterns affords not only a stereocomplementary approach to traditional organic synthesis, but also mandates a new perspective on the noncanonical *trans*-hydroboration mechanism. A combination of control experiments and computational studies at the DFT level of theory reveal the previously unrecognized role of the HMDS counteranion in a stepwise intermolecular hydrogen transfer process.

Boron-containing drug candidates have garnered increasing attention by virtue of their unique pharmacological properties and the growing prominence of boron neutron capture therapy (BNCT)[1–3]. As the example of 1,2-benzazaborine[4–8] serving as an alternative to naphthalene shows, the bioisosteric substitution of carbon by boron in the ring has emerged as a compelling strategy for the design of boracyclic frameworks and the advancement of drug discovery (Fig. 1A). In the design and exploitation of boron-containing drugs, B(OH)₂ often serves as a bioisosteric alternative to COOH because of the identical number of valence electrons, a comparable molecular geometry, and the ability to engage in hydrogen bond interactions with a biological receptor[9]. Analogously, oxaboroles, in their own right, emerge as promising

pharmacophores present in fungicidal molecules[10,11] comparable to butenolides that are prominent structural motifs featured in numerous antifungal natural products as well as drug molecules. Therefore, based on the concept of bioisosterism, we proposed that the five membered-ring hemiboronic esters represent potential bioisosteres of butenolides. Moreover, oxaboroles represent a potent scaffold for the stereospecific assembly of multi-substituted allylic alcohols, which constitute the core skeleton in many pharmaceuticals (Fig. 1B).

The classical hydroboration of alkynes has proven to be an exceptionally useful method for the stereospecific construction of linear alkenyl boronates due to its rigorous stereochemical course resulting from *cis*-addition of a B-H bond to the π-system[12–15]. During

[1]Jiangsu Key Laboratory of Drug Target Research and Drug Discovery of Neurodegenerative Disease, School of Pharmacy, Nanjing University of Chinese Medicine, Nanjing, China. [2]State Key Laboratory of Coordination Chemistry, Jiangsu Key Laboratory of Advanced Organic Materials, Chemistry and Biomedicine Innovation Center (ChemBIC), School of Chemistry and Chemical Engineering, Nanjing University, Nanjing, China. [3]Max-Planck-Institut für Kohlenforschung, Mülheim an der Ruhr, Germany. [4]These authors contributed equally: Yuan-Wen Liu, Yu Liu. ✉e-mail: jie.han@nju.edu.cn; fuerstner@kofo.mpg.de; hmjin@njucm.edu.cn

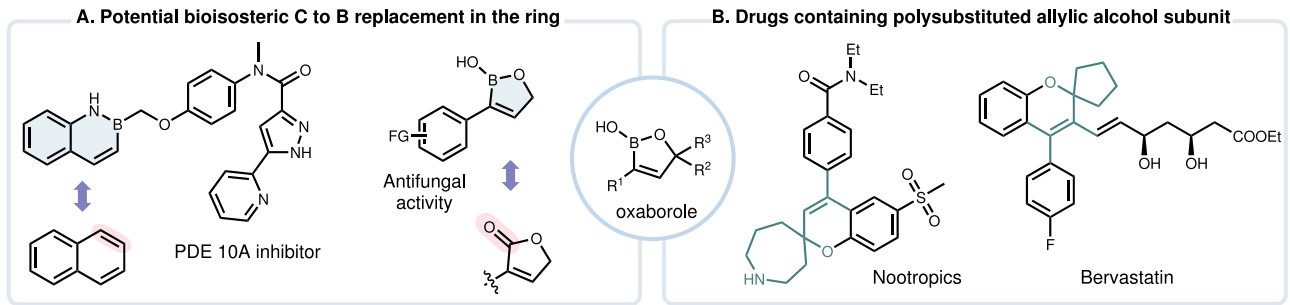

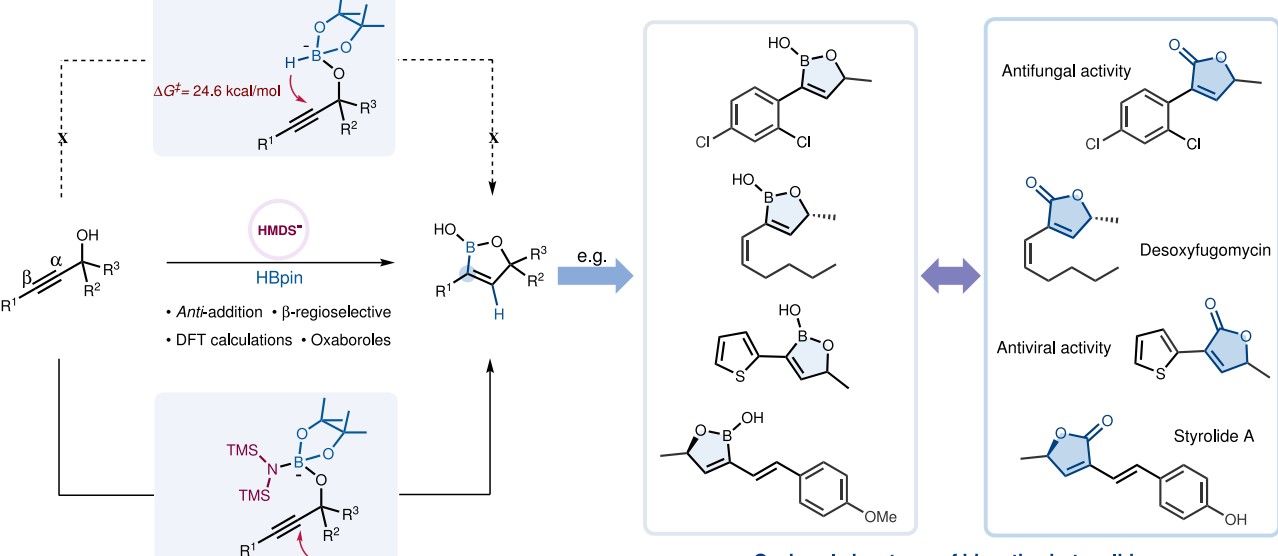

**Fig. 1 | Counteranion-enabled regioselective *anti*-addition of alkynes.**
**A** Exploring boron as a bioisosteric replacement for carbon in the ring.
**B** Polysubstituted allylic alcohol-containing drugs. **C** Synthesis of boracycles via

alkyne hydroboration. **D** *Trans*-Selective Chan alkyne reduction. **E** Chloride counteranion influence on Ru-catalyzed *trans*-hydrometallation. **F** Counteranion-enabled *trans*-selective hydroboration of alkynes.

the past decade, although considerable efforts were dedicated to exploring the stereo-complementary *anti*-addition mode[16–32], further expanding the potential of *trans*-hydroboration for the boracycle synthesis still remains underexplored (Fig. 1C)[33,34]. In this context, Chan alkyne reduction[35] with *trans*-stereoselectivity shows the in-situ formation of cycloaluminum species before quenching (Fig. 1D, left). This

successful way to challenge the canonical *syn*-selectivity consists in a *quasi*-intramolecular hydride delivery followed by trapping of the aluminum functionality from the rear. However, to date, there have been few reports on an analogous access to isolable oxaboroles by using benign boranes instead of aluminohydride reagents (Fig. 1D, right).

**Table 1 | Optimization of Reaction Conditions[a]**

| Entry | X equiv. base | Borane | Time (h) | Yield (%)[b] |
|---|---|---|---|---|
| 1 | 1.1 equiv. nBuLi | HBpin | 4 | 0 |
| 2 | 1.1 equiv. LDA | HBpin | 4 | 8 |
| 3 | 1.1 equiv. tBuOLi | HBpin | 4 | 0 |
| 4 | 1.1 equiv. LiHMDS | HBpin | 4 | 20 |
| 5 | 1.1 equiv. NaH | HBpin | 4 | < 5 |
| 6 | 1.1 equiv. NaNH$_2$ | HBpin | 4 | 0 |
| 7 | 1.1 equiv. NaHMDS | HBpin | 4 | 28 |
| 8 | 1.1 equiv. KHMDS | HBpin | 4 | 25 |
| 9 | 1.5 equiv. NaHMDS | HBpin | 4 | 65 |
| 10 | 2.0 equiv. NaHMDS | HBpin | 4 | 70 (64)[c] |
| **11** | **2.0 equiv. NaHMDS** | **HBpin** | **6** | **86** |
| 12 | 2.0 equiv. NaHMDS | HBpin | 12 | 85 |
| 13 | 2.0 equiv. NaHMDS | BH$_3$ | 6 | 73 |
| 14 | 2.0 equiv. NaHMDS | HBCat | 6 | 35 |
| 15 | 2.0 equiv. NaHMDS | 9-BBN | 6 | 0 |
| 16 | 2.0 equiv. NaHMDS | BH$_3$•SMe$_2$ | 6 | 0 |

[a]**1** (0.2 mmol), base and borane (0.4 mmol) in THF (2 ml) at room temperature, then quenching with NH$_4$Cl solution. [b]isolated yield. [c]1,4-dioxane as a solvent.

Metal counterions have proven capable of finely tuning reactivity and selectivity in metal catalysis through their synergistic interaction with substrates, steering intermediate formation[36–40]. Recently, ruthenium-catalyzed *trans*-hydrostannation as well as *trans*-hydrosilylation of propargyl alcohols have been developed that provide robust access to various α-functionalized trisubstituted allylic alcohols with exceptional levels of stereo- and regioselectivity (Fig. 1E, left)[41–45]. Assistance by the coordinating chloride ion unit of the [Cp*RuCl] catalyst able to engage in hydrogen bonding with an unprotected propargylic -OH group ensures this unorthodox reactivity[46,47]. However, this directing strategy cannot be extended to analogous *trans*-hydroboration reactions because the common borane reagents will instantly react with the hydroxy group. When suitably protected propargyl alcohol derivatives are used instead, without the hydrogen bonding direction, the *E/Z*- and regioisomer ratios in the ruthenium-catalyzed hydroboration proved to be quite variable and primarily dependent on steric factors of alkyne substrates (Fig. 1E, right)[48]. Hence, a broadly applicable method for the regiocontrolled *trans*-hydroboration of propargyl alcohols with further formation of valuable (*E*)-β-borocycles remains a highly desirable yet challenging task.

Outlined below (Fig. 1F), we disclose a stereo-complementary *trans*-selective addition of HBpin to alkynes promoted solely by hexamethyldisilazide (HMDS) anion. The reaction leverages the directing effect of readily available propargylic hydroxy groups and proceeds without any transition metal catalyst[49,50]. While previous studies on the alkali metal reagent-activated canonical *syn*-stereospecific addition of boranes to alkynes[51–53] were mostly focused on demonstrating the efficacy of the alkali metal cations, particularly that of Li(I)[54–56], the role of the counteranion has been largely overlooked or considered to be hardly relevant. A set of mechanistic control experiments in combination with DFT calculations suggest that a so-far unique anion-mediated stepwise *trans*-hydroboration process is operative. Distinct from the Chan's alcohol-aluminum adduct undergoing an

intramolecular B-H shift, a three component adduct intermediate involving the HMDS anion is generated and subjected to an intermolecular hydride transfer. The resulting oxaboroles are not only bioisosteric to butenolides, but also function as highly versatile building blocks for downstream functionalization to access a series of β-functionalized allylic alcohols.

## Results

The tertiary propargyl alcohol **1** was selected as model compound, which was reacted with 2 equivalents of HBpin and an alkali metal reagent in THF as the solvent; this choice was made because such highly sterically hindered substrates, in protected form, tend to undergo *cis*-addition when subjected to Ru-catalyzed hydroboration[48]. As depicted in Table 1, a series of lithium reagents was examined first, with LiHMDS producing the desired product, that is the cyclic boronate **2**, in 20% yield (entry 1-4). Other bases mainly induced the decomposition of **1** into phenylacetylene; this observation suggested that the [HMDS]⁻ anion plays a pivotal role for the reaction. In line with this notion, NaH as well as NaNH$_2$ proved unfruitful (entry 5, 6) whereas the use of NaHMDS or KHMDS both furnished **2** in similar, though modest, yield (entries 7 and 8). A significant improvement was reached by increasing the equivalents of base and extending the reaction time (entry 9-12). With 2 equivalents of NaHMDS, the yield was improved to 86% after 6 h (entry 11). Importantly, no side products formed by *cis*-addition were detected. Other borane reagents were evaluated as well. Simple BH$_3$ performed well with 73% yield (entry 13), catecholborane (HBCat) only led to 35% yield (entry 14), whereas 9-BBN and BH$_3$•SMe$_2$ did not give the product (entry 15 and 16). Finally, 2 equivalent of NaHMDS with HBpin turned out to be best suited.

### Substrate scope

With the optimized conditions established, we explored the substrate scope of this *trans*-hydroboration (Fig. 2). Tertiary

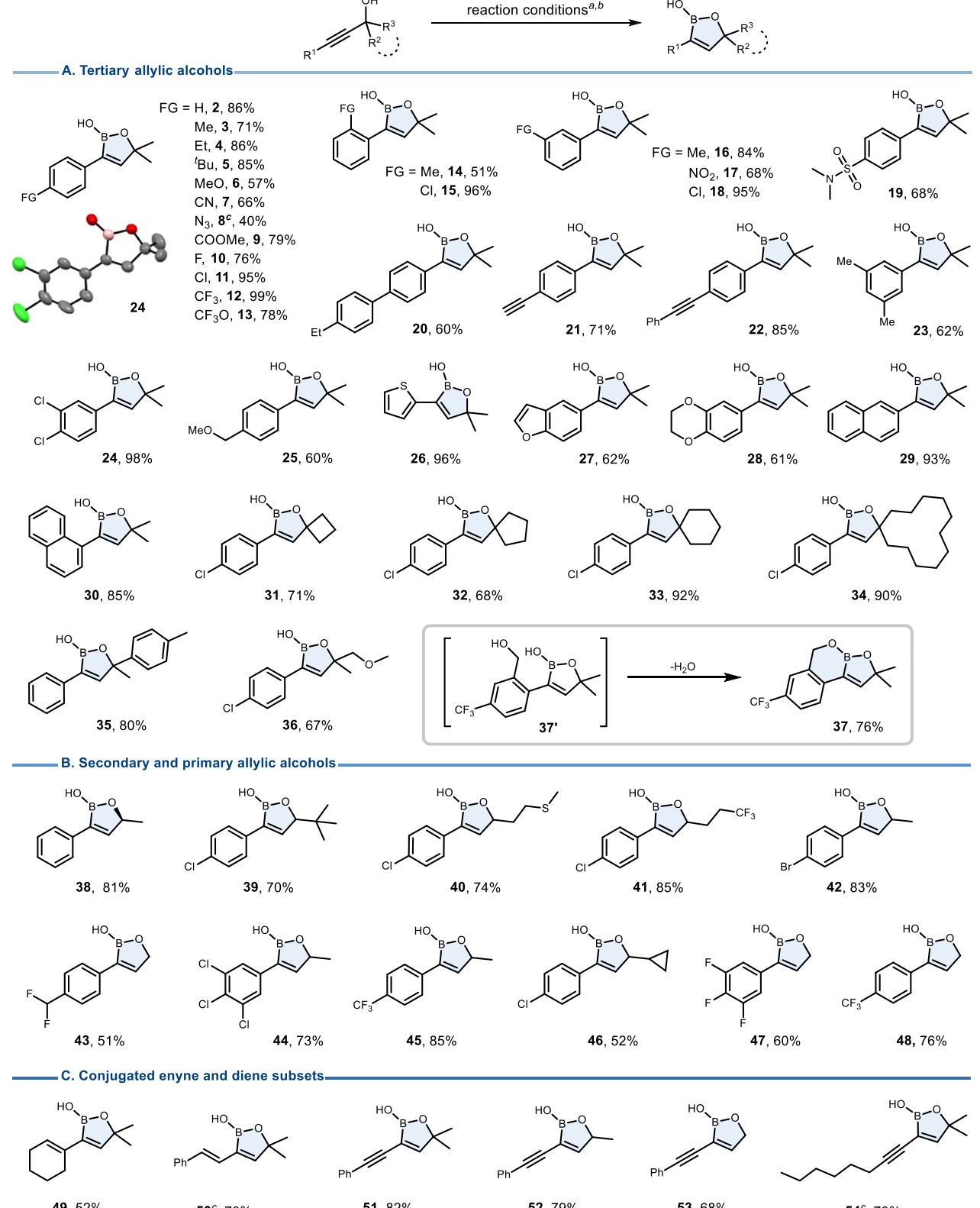

**Fig. 2 | Substrate scope.** [a]General reaction conditions: NaHMDS (0.4 mmol, 2 M in THF), propargyl alcohol (0.2 mmol), HBpin (0.4 mmol), THF (2 ml), room temperature, 6 h, then quenching with aqueous NH₄Cl solution. [b]Isolated yield. [c]NaHMDS (0.24 mmol), HBpin (0.2 mmol), 0 °C, 12 h. **A** The scope of tertiary allylic alcohols. **B** The scope of secondary and primary allylic alcohols. **C** The scope of conjugated enyne and diene subsets.

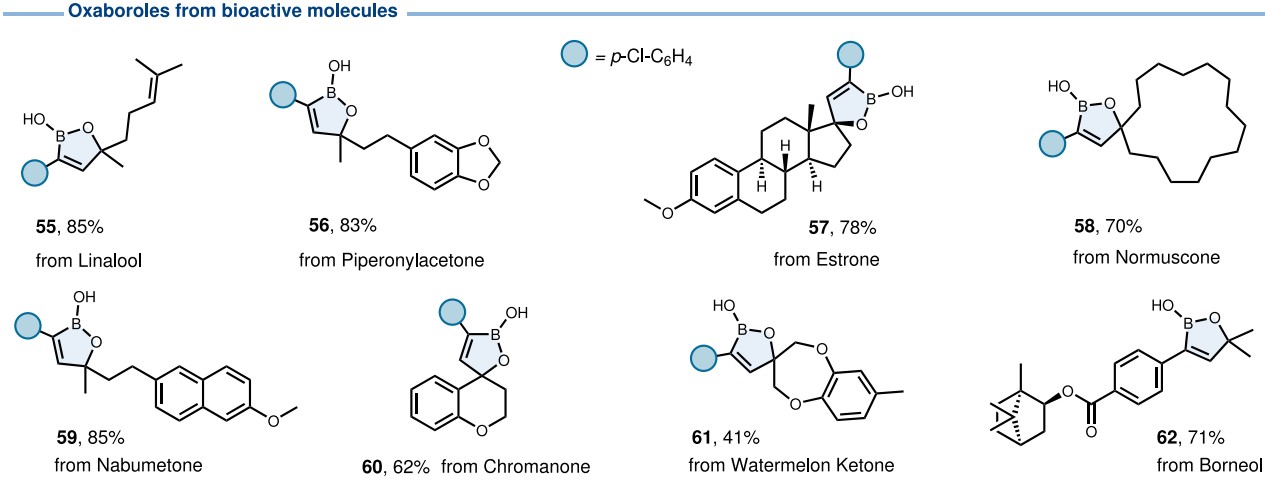

**Fig. 3 | Products of late-stage modifications of complex alkynes.** Oxaboroles derived from bioactive molecules.

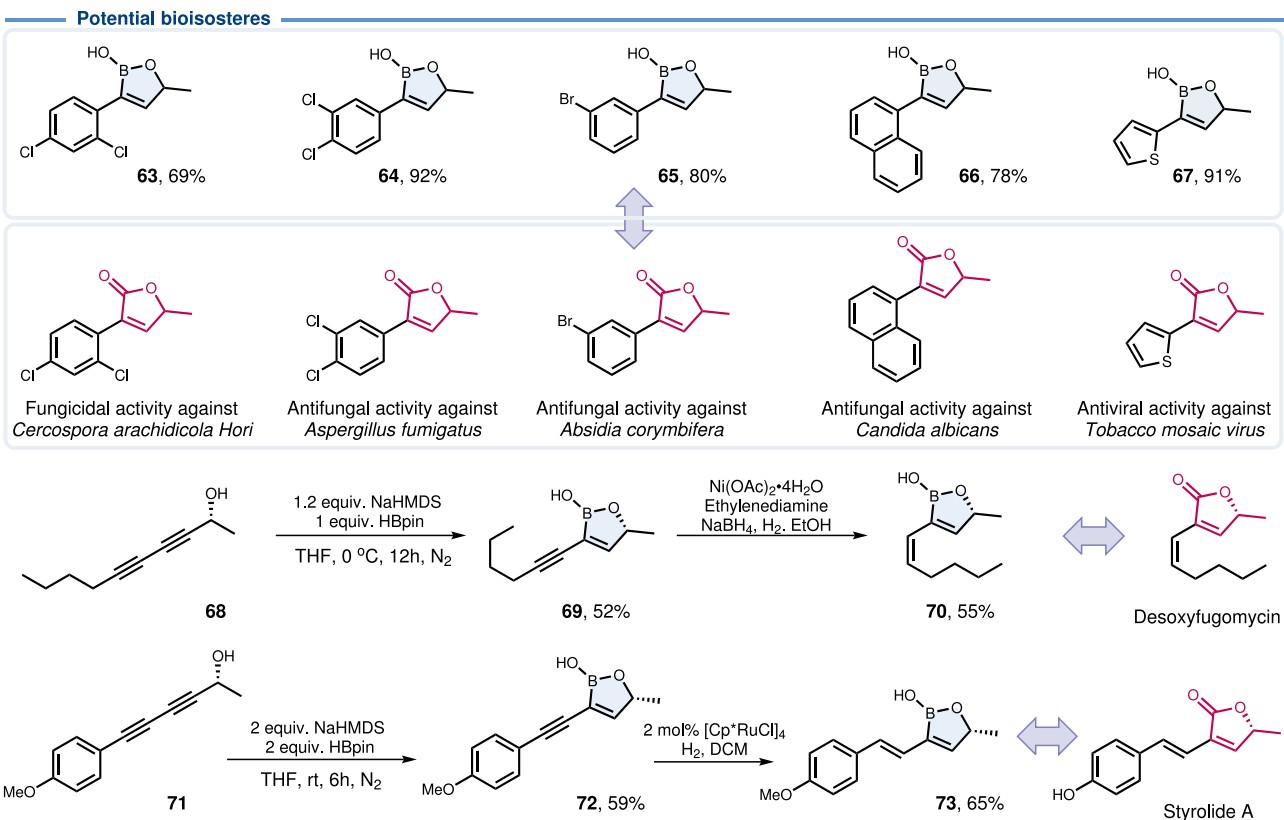

**Fig. 4 | Synthesis of oxaborole bioisosteres of butenolides.** Carbonyl-to-B(OH) bioisosteric replacement.

propargyl alcohols bearing different functional groups reacted smoothly to form the corresponding oxaborole products in good to excellent yields. Propargyl alcohols with aryl substituents bearing electron-withdrawing groups tend to give higher yields, while electron-donating groups on the arene led to somewhat more moderate outcomes. Notably, a vast array of unsaturated functional groups was preserved under the reductive conditions, including -CN (**7**), azide (**8**), ester (**9**), -NO$_2$ (**17**) and sulfonamide (**19**) groups. Halogen, -CF$_3$, -OCF$_3$ and heterocyclic substituents were tolerated as well (**10-13, 26-28**). Isolated carbon-carbon triple bonds without an adjacent hydroxy group were found to be inert, which is a particularly remarkable aspect of chemoselectivity (**21, 22**). This highlights the crucial role of the hydroxyl

group's negative inductive effect in propargyl alcohols, which increases the electrophilicity of the triple bond, enabling the reaction. Spirocyclic boronate products could be easily prepared from cyclic tertiary alcohols, whereby even a strained four-membered ring remained intact (**31**). In addition, a fused bora-cycle was accessible by virtue of intramolecular condensation of the boronate primarily formed with a neighboring benzylic hydroxy substituent (**37**). In addition to the tertiary alcohols, secondary and primary propargyl alcohols also performed well, delivering the desired products in generally good yields, while leaving the chiral center intact (**38-48**). Regioselective *trans*-addition to propargyl alcohols with conjugated enyne or diyne subunits turned out to be feasible (**49-54**); once again, the distal

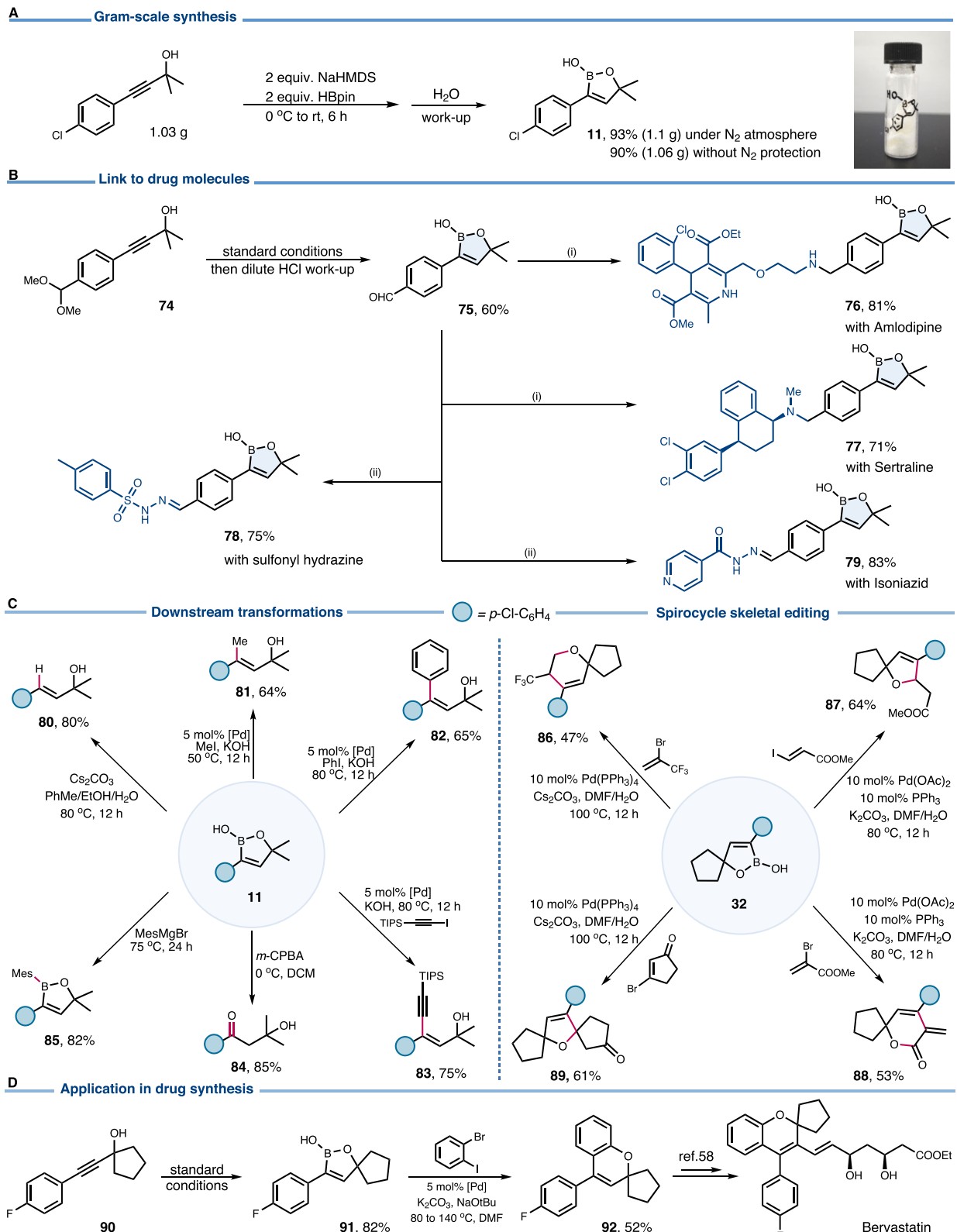

**Fig. 5 | Derivatizations and scale-up synthesis. A** Scale-up synthesis. **B** Decorations of drug molecules, (i) 1.2 equiv. related amine, 3 equiv. NaBH₃CN and 4 equiv. anhydrous MgSO₄. (ii) 1.2 equiv. hydrazide, anhydrous MeOH. [Pd] = PdCl₂(PPh₃)₂. **C** Boron-involved transformations. **D** Enabling streamlined drug synthesis.

alkene and alkyne units were shown not to interfere. However, alkyl substituted propargyl alcohols are currently not suitable for this reaction. The molecular structure assigned to **24** was confirmed by single-crystal X-ray diffraction (CCDC No.: 2346237).

The new methodology described herein represents a straightforward and productive approach to introduce the oxaborole motif into bioactive compounds, endowing the boron core unit with potential targeting ability (Figs. 3 and 4). Due to the good functional group compatibility, this five-membered

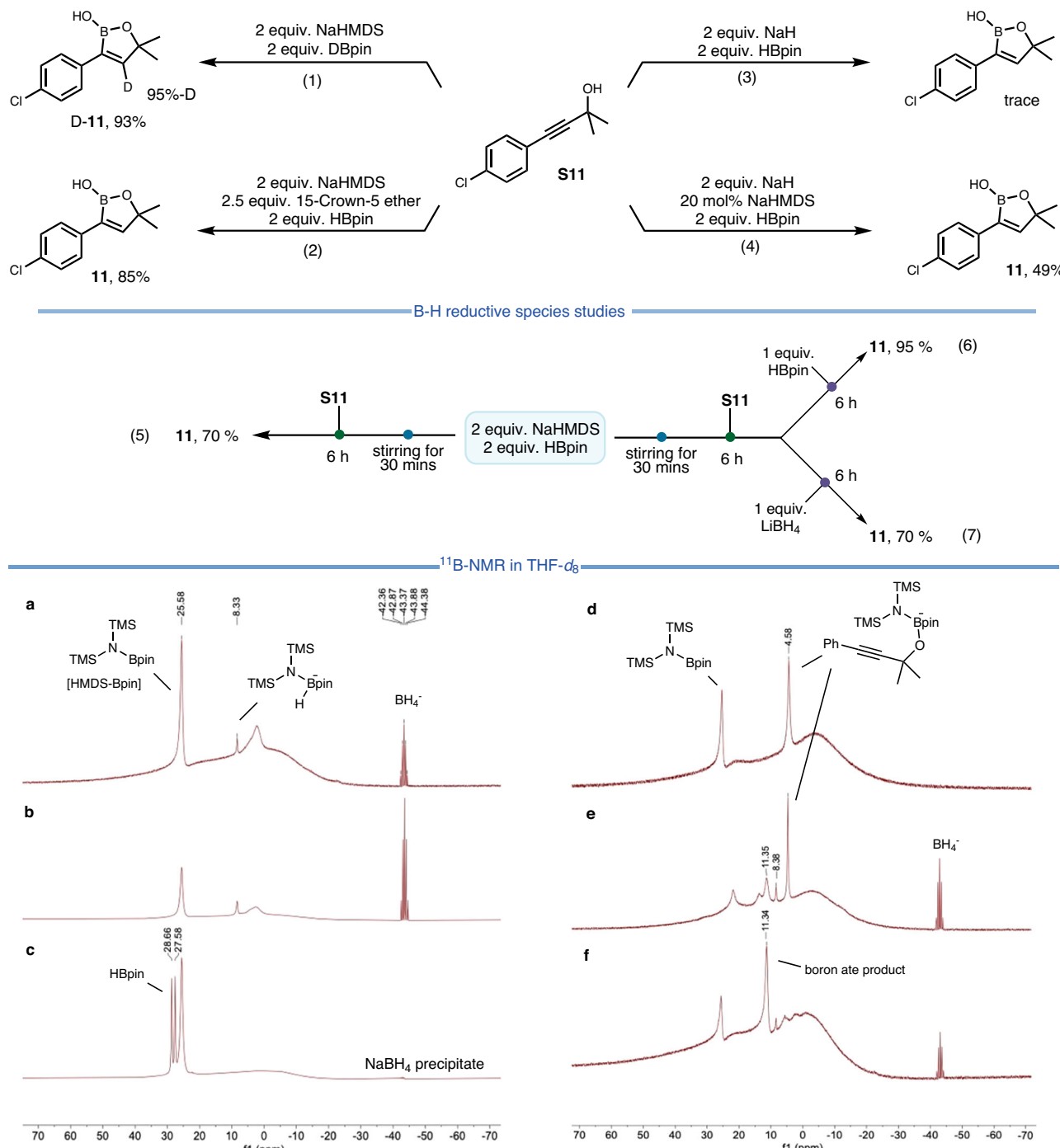

**Fig. 6 | Mechanistic experiments. a**, NaHMDS/HBpin = 1/2. **b**, NaHMDS/HBpin = 1/2.8. **c**, NaHMDS/HBpin = 1/3. **d**, Sodium alkoxide of **1** was added to freshly prepared [HMDS-Bpin]. **e**, Sodium alkoxide of **1** was added to a mixture of NaHMDS and HBpin (1:2). **f**, Standard condition after 6 h.

boracycle was successfully embedded into compounds derived from natural products or drug molecules upon their tethering to an alkyne, namely linalool (**55**), piperonylacetone (**56**), ethinyl-lestradiol (**57**), normuscone (**58**), nabumetone (**59**), chromanone (**60**) and watermelon ketone (**61**). Due to the good compatibility, the ester group could be adopted to tie borneol (**62**). Additionally, potential antifungal oxaboroles (**63-67**) were prepared to enrich the boron-based compound library. In general, an acidic environment may be more conducive to inhibiting fungal growth. Thus, they constitute potential bioisosteric equivalents of antifungal butenolide derivatives[57,58]. Two bora-butenolide isosterics

of bioactive natural products (**70** and **73**) were synthesized by further semi-hydrogenation.

### Derivatizations

To demonstrate the practicality of this NaHMDS-promoted *trans*-hydroboration, a gram-scale reaction was carried out, giving the target product in 93% yield. Even when carried out in air, a yield of 90% was obtained (Fig. 5A). A formyl group-bearing oxaborole building block was prepared (Fig. 5B, **75**), enabling facile installation of the oxaborole skeleton onto drug molecules at a late stage (**76** and **77**). The oxaborole moiety may enhance the antimicrobial activity in cooperation with

**Fig. 7 | Proposed mechanism.** Counteranion-involved *trans*-selective addition of B-H to alkynes.

hydrazide-hydrazones (**78** and **79**). This oxaborole was then subjected to diverse downstream transformations (Fig. 5C). Protodeborylation of **11** led to the formal *trans*-hydrogenation product **80**. A set of representative sp³-, sp²-, sp-hybridized carbon entities could be introduced at the β-position through Pd-catalyzed Suzuki-Miyaura cross coupling (**81-83**). Oxidative cleavage of C-B bond furnished the β-hydroxy ketone **84**. The B-OH group of **11** was replaced by a mesityl substituent on treatment with the corresponding Grignard reagent (**85**). Furthermore, versatile oxa-spirocyclic scaffolds are well within reach. This asset was exemplified by the synthesis of spirocyclic ether and spirolactone derivatives from **32** through tandem cross coupling/annulation (**86-89**). The oxaborole **91** was successfully employed on the stereoselective construction of the Bervastation framework (Fig. 5D)[59].

## Mechanistic investigations

Next, a series of control experiments was conducted to gain insights into the reaction mechanism (Fig. 6). Specifically, the hydride on the double bond was unambiguously shown to derive from the borane reagent by treatment of the propargyl alcohol substrate with DBpin, which resulted in 95% deuterium incorporation (eq. 1). To exclude any particular influence of the sodium ion, excess 15-crown-5 ether was added to the mixture; under these conditions, the product was formed in a largely unchanged 85% yield (eq. 2). In contrast, the use of NaH instead of NaHMDS produced only a trace amount of product (eq. 3). When NaH (2 equiv.) was supplemented with a substoichiometric amount of NaHMDS (20 mol%), however, a 49% yield was obtained (eq. 4).

These results underline the crucial role of the [HMDS]⁻ anion. Further insights were gained by ¹¹B-NMR spectroscopy, which revealed that HBpin is instantly transformed into [HMDS-Bpin] ($\delta = 25.6$ ppm)[60,61] and $BH_4^-$ (quint, $\delta = -43.4$ ppm) upon addition of NaHMDS (Fig. 6a); a small amount of the adduct derived from the [HMDS]⁻ ion

and HBpin was observed as well ($\delta = 8.3$ ppm) (see Figure S1 in the Supplementary Information). In contrast, when LiHMDS was added to HBpin in a 1:1 ratio, only a trace of HMDS-Bpin was detected in the ¹¹B-NMR spectrum, indicating that LiHMDS is ineffective in activating HBpin under standard conditions as compared to NaHMDS (see Figure S2 in the Supplementary Information). Moreover, the addition of KHMDS to HBpin in a 1:1 ratio resulted in the formation of a precipitate, showing that the intermediates with potassium counterion are not well soluble; this heterogenous system may hinder the reaction, finally leading to a low yield. ¹¹B-NMR titration experiments indicated that 1 equivalent of NaHMDS consumes approximately 2.8 equivalents of HBpin (Fig. 6b). When NaHMDS (0.2 mmol) was mixed with 3 equiv. HBpin in 0.5 ml THF, a mass of white precipitates was rapidly formed, which consists of insoluble $NaBH_4$ (Fig. 6c). When a mixture of NaHMDS and HBpin (1:1) was stirred for 30 min to ensure complete disappearance of HBpin before the propargyl alcohol was added, the reaction still proceeded well and furnished the product in 70% yield (eq. 5). In this case, a transient adduct formed by reaction of the alkoxide with [HMDS-Bpin] was detected by online ¹¹B-NMR ($\delta = 4.6$ ppm, Fig. 6e). When 1 equivalent of HBpin was added to the mixture of equation 5, the yield of **11** was increased to 95% (eq. 6); in contrast, addition of soluble $LiBH_4$ did not improve the yield (eq. 7). These results suggest that [HMDS-HBpin]⁻ or a related Na[B-H] species is the actual reducing agent delivering the hydride to the activated substrate, not $[BH_4]^-$. The product resulting from this *trans*-hydroboration reaction has a signal in the range of 11–13 ppm (Fig. 6f), which suggests that the species primarily formed is a tetrahedral boron-ate complex[62] prior to work-up.

On the basis of these experimental results, a stepwise *trans*-hydroboration mechanism is proposed that is triggered by NaHMDS (Fig. 7). Initially, the propargyl alkoxide **A** is generated at the cost of 1 equivalent of NaHMDS. The remaining NaHMDS promotes the

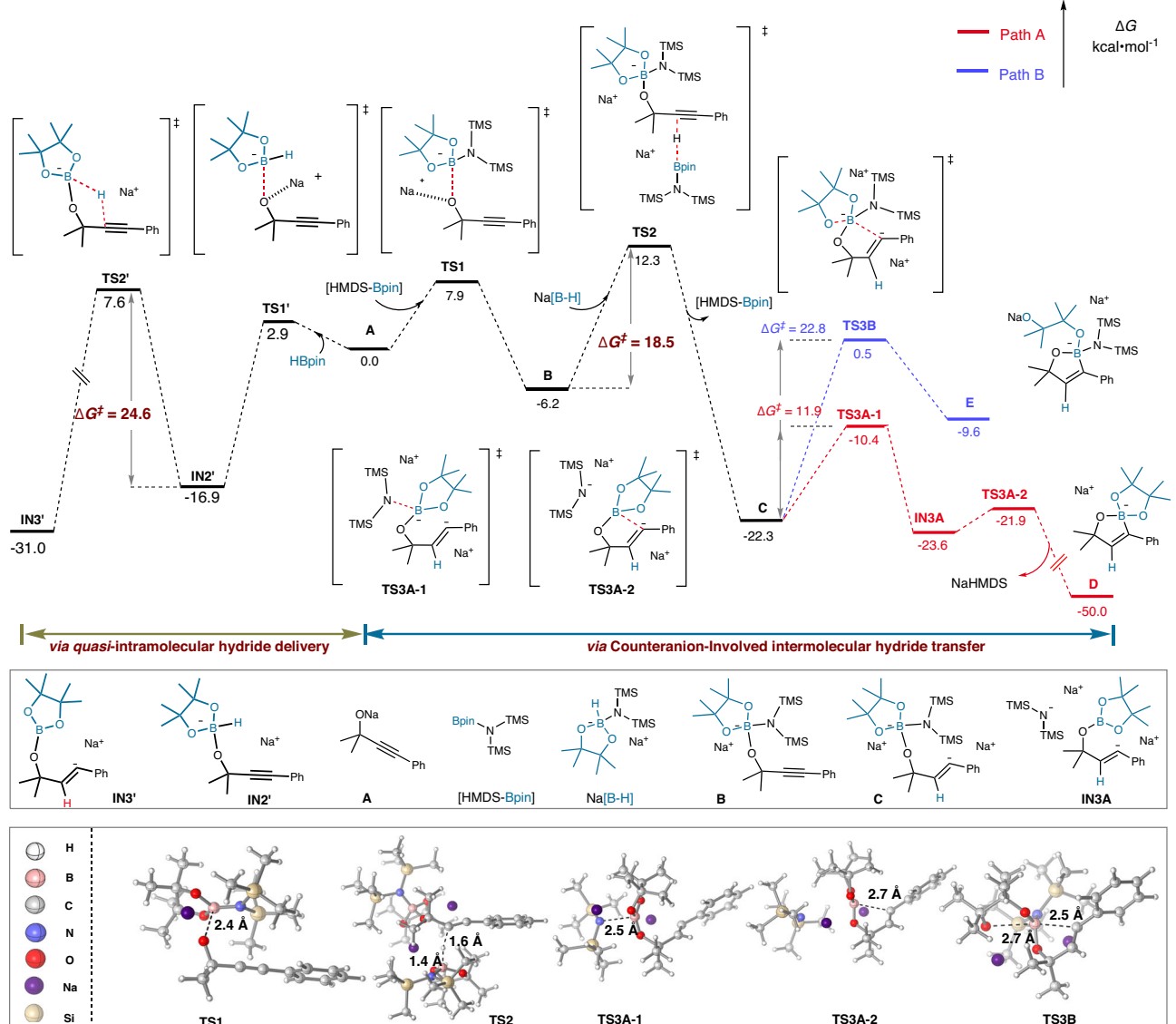

**Fig. 8 | Computational mechanistic study on the plausible mechanism.** Free-energy profiles of reaction pathways computed at the DFT/B3LYP(D3BJ)/def2-SVP//M06-2X/def2-TZVPP/SMD (THF) level of theory. The geometries of the transition states are illustrated below, with the key bond lengths indicated in Angstrom (Å).

conversion of HBpin into Na[HMDS-HBpin], which may undergo disproportionation to form [HMDS-Bpin], NaBH₄ and other boron-ate complexes. The propargyl alkoxide **A** then adds to [HMDS-Bpin] to form the boron-ate complex **B**, which is reduced to give intermediate **C** via intermolecular hydride delivery. Due to the size of the HMDS ligand, the *cis*-configured isomer **C'** is disfavored on steric grounds. Subsequent boron capture with dissociation of an *N*- or *O*-ligand (path A and B) leads to the corresponding boron-ate complexes **D** and **E**, respectively, which are hydrolyzed upon aqueous work-up to yield the final oxaborole product **2**.

## Computational studies

To gain insight into whether pathway A or B is operative and to elucidate the reaction mechanism of the *trans*-hydroboration process in more detail, we performed DFT calculations for the model reaction (Fig. 8). Based on the aforementioned experimental results, we took the propargyl alkoxide (**A**) as the starting point. The reaction initiates with the coupling between **A** and the [HMDS-Bpin] species, generating a boron-ate complex **B** through the transition state **TS1**, with an energy barrier (Δ*G*‡) of 7.9 kcal/mol. Subsequently, Na[B-H] attacks the alkyne group of **B** to complete an intermolecular hydride transfer via **TS2**

(Δ*G*‡ = 18.5 kcal/mol), generating intermediate **C**. The reaction pathway diverges at **C**, yielding the target product **2** through intermediate **D** (path A) or intermediate **E** (path B). Despite the negative charge of intermediate **C**, charge population analysis shows that the boron atom itself is actually positively charged due to electron withdrawal by the adjacent electronegative *N*- and *O*-ligands (see Figure S3 in the Supplementary Information). In path A, **C** first overcomes a transition state **TS3A-1** to break the B-N bond, forming intermediate **INT3A**. The negatively charged carbon atom bonded to the phenyl group in **C** then attacks the boron atom, passing through the second transition state **TS3A-2**. This leads to the formation of the cyclized intermediate **D** with simultaneous removal of NaHMDS. The final oxaborole product **2** is then obtained upon hydrolysis of **D**. Path A exhibits an overall energy barrier of 11.9 kcal/mol ($G_{(TS3A-1)}$-$G_{(C)}$), and the product formation is exothermic by 27.7 kcal/mol ($G_{(D)}$-$G_{(C)}$). Conversely, path B involves breaking of the B-O bond through a single transition state **TS3B** and simultaneous formation of another cyclized intermediate **E** with a substantially higher Δ*G*‡ value of 22.8 kcal/mol ($G_{(TS3B)}$-$G_{(C)}$) and Δ*G* value of 12.7 kcal/mol ($G_{(E)}$-$G_{(C)}$). Thus, path A is both kinetically and thermodynamically more favorable than path B. On the other hand, intermediate **C** can convert to its *cis*-configured isomer **C'** with a free

energy change of −1.1 kcal/mol. However, due to the steric hindrance exerted by the TMS group, the formation of the cyclized *cis*-intermediate is infeasible, with a free energy barrier as high as 37.2 kcal/mol (see Figure S4 in the Supplementary Information). Therefore, it can be concluded that the *trans*-hydroboration reaction predominantly follows path A, which is more favorable compared to the *cis*-hydroboration reaction via path A′. In addition, an intramolecular hydride shift without the assistance of HMDS anion was calculated (Fig. 8, left), revealing a higher energy barrier that hinders this process. ($\Delta G^{\ddagger} = 24.6$ kcal/mol). Thus, the critical role of the [HMDS]$^{-}$ anion is attributable to its electronegative nitrogen atom and the steric hindrance of the TMS groups, which together facilitate the B-H transfer, B-C bond formation and cyclization, and impose the observed stereoselective course. We have also explored the mechanism of a possible direct hydroboration reaction between sodium alkoxide **A** and Na[HBpin·HMDS] (Na[B-H]). However, DFT calculations suggest that such a mechanism is energetically unfeasible (see Figure S5 in the Supplementary Information).

## Discussion

We report a transition metal-free, counteranion-enabled *trans*-hydroboration of alkynes, which features mild reaction conditions, a broad substrate scope, a remarkable functional group tolerance, and excellent regioselectivity. This transformation is readily scalable and enables a modular and robust access to the oxaborole scaffold. Because of the growing importance of boron-containing drugs, the relevance of the method was illustrated by applications to bioactive molecules as well as by the construction of bioisosteres of butenolides. Moreover, the oxaborole products constitute a flexible platform for the assembly of various trisubstituted alkenes and *oxa*-spirocycles. Experimental and computational studies at the DFT level of theory suggest that the reaction proceeds by an anion-mediated stepwise *anti*-addition process, whereby the [HMDS]$^{-}$ counterion accounts for the generation of the active hydride species as well as the formation of the reducible boron-ate intermediate.

## Methods

### General procedure via 2 equiv. HBpin

An oven-dried Schlenk tube equipped with a magnetic stir bar was charged with propargyl alcohol (0.2 mmol), anhydrous THF (2 mL) and NaHMDS (2.0 M THF solution, 0.4 mmol) at 0 °C under N$_2$ atmosphere. The mixture was stirred for 10 min at 0 °C, and then HBpin (0.4 mmol, 58 µL) was introduced. Subsequently, the reaction vessel was sealed with a Teflon-lined screw cap and the mixture warmed to room temperature. After continuously stirring for 6 h, the reaction was quenched with aqueous NH$_4$Cl solution (10 mL), followed by extraction with EtOAc (3 × 10 mL). The combined organic layer was dried over anhydrous Na$_2$SO$_4$, filtered, and concentrated in vacuo. The residue was purified by flash chromatography on silica gel to afford the desired product.

### General procedure via 1 equiv. HBpin

An oven-dried Schlenk tube equipped with a magnetic stir bar was charged with propargyl alcohol (0.2 mmol), anhydrous THF (2 mL) and NaHMDS (2.0 M THF solution, 0.24 mmol) at 0 °C under N$_2$ atmosphere. The mixture was stirred for 10 min at 0 °C, then HBpin (0.2 mmol, 29 µL) was introduced. Subsequently, the reaction vessel was sealed with a Teflon-lined and kept in an ice bath. After continuously stirring for 12 h at 0 °C, the reaction was quenched with aqueous NH$_4$Cl solution (10 mL), followed by extraction with EtOAc (3 × 10 mL). The combined organic layer was dried over anhydrous Na$_2$SO$_4$, filtered, and concentrated in vacuo. The residue was purified by flash chromatography on silica gel to afford the desired product.

## Data availability

The X-ray crystallographic coordinates for structures reported in this study have been deposited in the Cambridge Crystallographic Data Centre under accession (CCDC), under deposition numbers 2346237. These data can be obtained free of charge from The Cambridge Crystallographic Data Centre via www.ccdc.cam.ac.uk/data_request/cif. All other data that support the findings of this paper, including starting material preparation, experimental procedures, compound characterizations, NMR spectra of new compounds, and DFT calculations, are available within the paper and its Supplementary Information. All data are also available from the corresponding author upon request. Source data are provided with this paper.

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

## Acknowledgements

H.J. are grateful for financial support from Natural Science Foundation of China (Grant No. 22201136), Science Foundation of Jiangsu Province (Grant No. BK20220463), Jiangsu Specially Appointed Professor Plan, Nanjing University of Chinese Medicine (Grant No. XPT22201136 and ZYXPY2024-005). J.H. acknowledges Natural Science Foundation of China (Grant No. 22101130) as well as the support from Xiaomi foundation. All theoretical calculations were performed at the High-Performance Computing Center (HPCC) of Nanjing University.

## Author contributions

H.J. conceived the work and designed the experiments. Y.-W.L. conducted the experiments. Y.L. completed density functional theory (DFT) calculations. Y.Z., M.Z., M.-E.R. and P.H. assisted the work. J.H. directed DFT calculations. A.F. and H.J. discussed and co-wrote the manuscript.

## Funding

## Competing interests

The authors declare no competing interests.
