## [Transparent Peer Review file · Nature Communications]

Expedient access to bora-butenolide bioisosteres by counteranion-mediated trans-hydroboration of alkynes

Corresponding Author: Professor Hongming Jin

Version 0:

Reviewer comments:

Reviewer #1

(Remarks to the Author)

This manuscript by Han, Fürstner, Jin, and coworkers describes an efficient synthesis of oxaboroles as butanolide bioisoster via trans-hydroboration of propargylalcohols. The substrate scope and yields are reasonable. The synthesis of potential bioisosters and the derivatizations of obtained products are also well-investigated. However, the reported method, unfortunately, lacks significant novelty because this strategy (a combination of propargylalcohol, base, and boron reagents) has already been extensively established, affording more high-valued oxaborole derivatives than the present reported products. While the work demonstrates a wide variety of oxaborole syntheses, the differences between the present work and the previous works (e.g., ref. 48) are only a boron reagent (HBpin vs. B2pin2) and a base (NaHMDS vs. BuLi), which does not sufficiently advance beyond existing methodologies to meet the high novelty and impact standards required for Nature Communications. In addition, given the prior disclosures on various transformations of the obtained oxaboroles, the reuse of these methods in the current study does not represent a novel contribution. The proposed mechanistic cycle is intriguing and represents a new pathway in trans-hydroboration reactions. However, as a result, the present mechanism should require an excess amount of NaHMDS and HBpin, which may weaken its synthetic value. Overall, while this work provides good synthetic examples and mechanistic insights, I do not support this work in Nature Communications.

Reviewer #2

(Remarks to the Author)

This manuscript describes the development of an interesting method for the synthesis of bora-butenolide derivatives. I am sure that this method is synthetically very useful because the reaction can be performed using quite a simple procedure and only commercially available and inexpensive reagents (HBpin and NaHMDS). The starting propargyl alcohols are also easily available, and the obtained oxaboroles are potential products as biologically active compounds. Indeed, a large number of the substrates are shown in Table 2, and impressive applications such as large-scale synthesis and post-transformation of oxaboroles (Figure 3) would attract the interest of many readers. Mechanistic studies performed experimentally and computationally in Figures 4 and 6 should help readers understand the proposed mechanism. Overall, I think that this manuscript is well written and therefore deserves publication in Nat. Commun. after minor revisions.

Suggested revisions:

1) The authors use the term "bioisostere" in this manuscript. As cited in ref.9, oxaborole derivatives are known to exhibit interesting biological activity. However, would oxaborole derivatives actually function as bioisosteres of butenolides? If yes, please show any evidence in the introduction.

2) As shown in Table 1, the effect of an Na ion is crucial for the reaction. Why is the product yield low when Li and K ions are involved in the reaction? Please add any comment in the text.

3) The reaction mechanism is interesting. If NaHMDS works as a catalyst in the process of the formation of a borole, 1.1 equiv of NaHMDS should provide the product in good yield. Why does the reaction require 2 equiv of NaHMDS?

4) In Figure 5, the process to form Bpin-HMDS from NaHMDS and HBpin might mislead readers. The resultant Na[HBpin-HMDS] would undergo a disproportionation reaction to form Bpin-HMDS and a mixture of borohydride. On the other hand, the reaction between Na[HBpin-HMDS] and intermediate B also provides Bpin-HMDS. Please clarify this issue.

5) How could the authors eliminate a possibility of the hydroboration reaction between a sodium alkoxide A and Na[HBpin-HMDS] to directly form intermediate C?

Reviewer #3

(Remarks to the Author)

In this manuscript Han, Furstner, Jin, and coworkers reported the trans-hydroboration of propargylic alcohols which is mediated by NaHMDS. A large number of oxaboroles was synthesized by using HBpin as the borylating reagent. The scope and limitation, as well as the derivatization of the oxaboroles, have been studied thoroughly. The mechanism of this attractive reaction was studied by control experiments and DFT calculations. The reaction was assumed to proceed via intermolecular hydride transfer from a borohydride species to alkyne.

The study is well done and it will provide a new, practical method for the synthesis of oxaboroles. The proposed mechanism is very interesting, but I have some questions related to the mechanism. I recommend this manuscript for publication provided that the following issues are addressed.

1. The study disclosed that simple alkynes did not react with NaHMDS-HBpin, while propargyl alcohols were reactive. The reason for this result should be discussed.

2. In the DFT calculations, the sodium ion occasionally was occasionally included in the intermediate/transition state. In the control experiment, the authors concluded that that the influence of sodium ion is very small. Please explain.

Reviewer #4

(Remarks to the Author)

This work realizes a synthesis of five-membered boracycles with moderate to good yields through the anti-addition of pinacolborane to alkynes, catalyzed by the HMDS counteranion. The manuscript demonstrates that [HMDS-HBpin]⁻ is a key reducing agent in the anti-addition process, as confirmed by various control experiments. Additionally, the authors explore the mechanism behind the preference for the anti-addition pathway and provide an energy profile of the proposed reaction route using DFT calculations. The findings indicate that the steric hindrance from the TMS group results in a higher free energy barrier for the cis-addition process compared to the anti-addition process, explaining the observed preference for the latter when pinacolborane reacts with alkynes. From an application standpoint, the ability to conduct the reaction on a gram scale and the variety of downstream transformations highlight the potential utility of this boracyclic framework construction strategy. Overall, this work is innovative, and I believe the manuscript could be appropriate for publication in Nat. Commun. after addressing the following points:

1. In Fig. 4, the ratio of NaHMDS to HBpin appears inconsistent with the experimental data in the SI. In the section titled "Representative procedure for the NMR titration experiment, Fig. 4a-c" the authors do not specify the concentration of HBpin, making it unclear how to determine the NaHMDS/HBpin ratio in Fig. 4a-c. Furthermore, in the "Representative procedure for Fig. 4e" a mixture of NaHMDS (2 M in THF, 0.2 mmol, 100 μ L) and HBpin (0.2 mmol, 30 μ L) was used, resulting in a NaHMDS/HBpin ratio of 1:1, rather than the 1:2 indicated in Fig. 4e.

Version 1:

Reviewer comments:

Reviewer #2

(Remarks to the Author)

This reviewer agrees that the authors properly addressed comments raised by all reviewers. The revised mechanism discussion would fully convince readers. I recommend that the authors incorporate part of their response about "bioisostere" to Comment 1 from Reviewer #2 into the revised manuscript. After this very minor revision, I fully agree with publication of this manuscript in Nat. Commun.

Reviewer #3

(Remarks to the Author)

The authors responded and modified the manuscript based on the comments of the reviewers appropriately, and now I recommend this manuscript for publication.

Reviewer #4

(Remarks to the Author)

I have thoroughly examined the submission of this manuscript and am pleased to report that the authors have diligently revised it, I am now very satisfied and the current format will be acceptable to me.

Response to the reviewers

Reviewer #1 (Remarks to the Author):

This manuscript by Han, Fürstner, Jin, and coworkers describes an efficient synthesis of oxaboroles as butanolide bioisoster via *trans*-hydroboration of propargylalcohols. The substrate scope and yields are reasonable. The synthesis of potential bioisosters and the derivatizations of obtained products are also well-investigated. However, the reported method, unfortunately, lacks significant novelty because this strategy (a combination of propargylalcohol, base, and boron reagents) has already been extensively established, affording more high-valued oxaborole derivatives than the present reported products. While the work demonstrates a wide variety of oxaborole syntheses, the differences between the present work and the previous works (e.g., ref. 48) are only a boron reagent (HBpin vs. B₂pin₂) and a base (NaHMDS vs. BuLi), which does not sufficiently advance beyond existing methodologies to meet the high novelty and impact standards required for Nature Communications. In addition, given the prior disclosures on various transformations of the obtained oxaboroles, the reuse of these methods in the current study does not represent a novel contribution. The proposed mechanistic cycle is intriguing and represents a new pathway in *trans*-hydroboration reactions. However, as a result, the present mechanism should require an excess amount of NaHMDS and HBpin, which may weaken its synthetic value. Overall, while this work provides good synthetic examples and mechanistic insights, I do not support this work in Nature Communications.

Author's reply: Thanks for your comments. The n-BuLi-promoted *trans*-borylative strategy of propargyl alcohols, developed by Prof. Uchiyama and his co-workers, emerged as a powerful tool for constructing fully substituted oxaborole derivatives. However, its generality is limited by the narrow scope with regard to the boron reagents in that only B₂Pin₂ and alkynyl boronate have been shown to be suitable (ref. 48 and 49). As correctly noted by the reviewer, B₂Pin₂ leads to *trans*-diboration products. Notably, however, any subsequent deboronative functionalization of these diboration products results in boracyclic opening because the reactivity of the boron within the ring is higher than that of the Bpin substituent on the ring (ref. 61). Thus, tri-substituted oxaboroles, as reported in the present manuscript, cannot be synthesized using the *trans*-diboration strategy followed by protodeboration. Our new *trans*-hydroboration protocol effectively addresses this synthetic challenge and is hence complementary to existing methodologies (see the figure below).

More significantly, in contrast to the *trans*-borylation reactions reported in literature, which proceed through a lithium cation-promoted intramolecular migration step, the mechanism of our new *trans*-hydroboration involves an anion-mediated intermolecular pathway. Moreover, our investigations revealed a previously unrecognized and fundamentally novel counteranion effect. Therefore, the differences between the present study and previous work clearly extend beyond the choice of boron reagent and base. HBpin and NaHMDS are commercially available and inexpensive reagents. The new transformation proceeds under mild conditions and does not need the high temperature required for n-BuLi-

mediated borylative processes. The scalability of this reaction and its broad tolerance for diverse functional groups demonstrate its synthetic utility, as explicitly mentioned by all other reviewers.

Reviewer #2 (Remarks to the Author):

This manuscript describes the development of an interesting method for the synthesis of bora-butenolide derivatives. I am sure that this method is synthetically very useful because the reaction can be performed using quite a simple procedure and only commercially available and inexpensive reagents (HBpin and NaHMDS). The starting propargyl alcohols are also easily available, and the obtained oxaboroles are potential products as biologically active compounds. Indeed, a large number of the substrates are shown in Table 2, and impressive applications such as large-scale synthesis and post-transformation of oxaboroles (Figure 3) would attract the interest of many readers. Mechanistic studies performed experimentally and computationally in Figures 4 and 6 should help readers understand the proposed mechanism. Overall, I think that this manuscript is well written and therefore deserves publication in *Nat. Commun.* after minor revisions.

Suggested revisions:

1) The authors use the term "bioisostere" in this manuscript. As cited in ref.9, oxaborole derivatives are known to exhibit interesting biological activity. However, would oxaborole derivatives actually function as bioisosteres of butenolides? If yes, please show any evidence in the introduction.

Author's reply: Thank you very much for your suggestion. In the design and exploitation of boron-containing drugs, B(OH)₂ often serves as a bioisosteric alternative to COOH due to the identical number of valence electrons, a comparable molecular geometry and the resultant similar ability to engage in hydrogen bond interactions with a biological receptor (see below, *J. Med. Chem.* **2021**, *64*, 17706–17727).

[FIGURE REDACTED]

Analogously, the literature also reports that certain oxaboroles exhibit antifungal bioactivity comparable to that of certain butenolides (ref. 9). Therefore, based on the concept of bioisosterism, we proposed that the five membered-ring hemiboronic esters represent potential bioisosteres of butenolides.

2) As shown in Table 1, the effect of an Na ion is crucial for the reaction. Why is the product yield low when Li and K ions are involved in the reaction? Please add any comment in the text.

Author's reply: Thank you very much for your suggestion. In contrast, when LiHMDS was added to HBpin in a 1:1 ratio, only a trace of HMDS-Bpin was detected in the ^{11}B -NMR spectrum, indicating that LiHMDS is ineffective in activating HBpin under standard conditions as compared to NaHMDS (see Figure S2 in the Supplementary Information). Moreover, the addition of KHMDS to HBpin in a 1:1 ratio resulted in the formation of a precipitate, showing that the intermediates with potassium counterions are not well soluble; this heterogenous system may hinder the reaction, finally leading to a low yield. We added these explanations in the section of the revised manuscript describing the mechanistic investigations.

Figure S2. The ^{11}B NMR spectrum of LiHMDS/HBpin (1:1) compared with NaHMDS/HBpin (1:1) ($\text{THF}-d_8$).

3) The reaction mechanism is interesting. If NaHMDS works as a catalyst in the process of the formation of a borole, 1.1 equiv of NaHMDS should provide the product in good yield. Why does the reaction require 2 equiv of NaHMDS?

Author's reply: Thank you very much for the good question. According to the online ^{11}B -NMR, the reaction of HBpin and NaHMDS not only generates $[\text{HMDS-HBpin}]^-$, but also produces $[\text{BH}_4]^-$ and some boron-ate complexes ($\delta = 1\text{-}5$ ppm); some of the NaHMDS is hence consumed by the inevitable side reactions. The control experiments indicated that NaBH_4 cannot reduce the triple bond of propargyl alcohols. Thus, the use of only 1.1 equiv. of NaHMDS results in a low yield. To prevent readers from misinterpreting it as a catalytic cycle, we have redrawn the mechanism in the revised Fig. 5.

4) In Figure 5, the process to form Bpin-HMDS from NaHMDS and HBpin might mislead readers. The resultant $\text{Na}[\text{HBpin-HMDS}]$ would undergo a disproportionation reaction to form Bpin-HMDS and a mixture of borohydride. On the other hand, the reaction between $\text{Na}[\text{HBpin-HMDS}]$ and intermediate B also provides Bpin-HMDS. Please clarify this issue.

Author's reply: Thank you very much for your suggestion. We have redrawn the mechanism to indicate the two access routes to HMDS-Bpin, see the revised Fig. 5.

5) How could the authors eliminate a possibility of the hydroboration reaction between a sodium alkoxide A and $\text{Na}[\text{HBpin-HMDS}]$ to directly form intermediate C?

Author's reply: Thank you very much for your insightful comment. We have carefully evaluated this possibility through additional computational analysis, and we have included the relevant discussion in both the revised manuscript and the Supplementary Information. Our DFT calculations indicate that the direct hydroboration reaction between sodium alkoxide A and $\text{Na}[\text{HBpin-HMDS}]$ (denoted as $\text{Na}[\text{B-H}]$) is not energetically feasible. We are grateful for your suggestion, which allowed us to strengthen the mechanistic discussion in the revised version.

To further clarify our findings, we have added the following statement to the revised manuscript: We have also explored the mechanism of a possible direct hydroboration reaction between sodium alkoxide A and $\text{Na}[\text{HBpin-HMDS}]$ ($\text{Na}[\text{B-H}]$). However, DFT calculations suggest that such a mechanism is energetically unfeasible (see Figure S5 in the Supplementary Information).

Corresponding details are provided in the revised Supplementary Information, as shown below:

We examined two mechanistic pathways for the direct hydroboration reaction between sodium alkoxide A and $\text{Na}[\text{HBpin-HMDS}]$ ($\text{Na}[\text{B-H}]$) (Figure S5). In path 1, $\text{Na}[\text{B-H}]$ initially attacks the alkyne moiety of A, initiating intermolecular hydride transfer via transition state TSA' . This step requires a high free activation energy ($\Delta G^\ddagger = 32.1$ kcal/mol), leading to the formation of intermediate INTA' . In the subsequent step, INTA' couples with the $[\text{HMDS-}$

[Bpin] fragment to form the boronate complex **C'** through transition state **TSB'** ($\Delta G^\ddagger = 9.1$ kcal/mol). While the second step is energetically accessible, the overall transformation is rendered unfavorable due to the prohibitively high barrier of the initial hydride transfer. In path 2, Na[B-H] first reacts with **A** via transition state **TSA''**, involving B–O bond cleavage and generating intermediate **INA''**. This step also requires a high activation energy ($\Delta G^\ddagger = 30.7$ kcal/mol). Subsequently, an intramolecular hydride transfer from boron to the alkyne moiety occurs through transition state **TSB''**, forming intermediate **C''**. The high barrier of the initial step again makes this pathway unlikely under the reaction conditions. In summary, both mechanistic possibilities entail free energy barriers exceeding 30 kcal/mol, suggesting that direct hydroboration is not a viable pathway under the reaction conditions employed.

Figure S5. Free energy profiles of direct hydroboration reaction between sodium alkoxide **A** and Na[HBpin-HMDS] (Na[B-H]), computed at the DFT/B3LYP(D3BJ)/def2-SVP//M06-2X/def2-TZVPP/SMD(THF) level of theory.

Reviewer #3 (Remarks to the Author):

In this manuscript Han, Furstner, Jin, and coworkers reported the trans-hydroboration of propargylic alcohols which is mediated by NaHMDS. A large number of oxaboroles was synthesized by using HBpin as the borylating reagent. The scope and limitation, as well as the derivatization of the oxaboroles, have been studied thoroughly. The mechanism of this attractive reaction was studied by control experiments and DFT calculations. The reaction was assumed to proceed via intermolecular hydride transfer from a borohydride species to alkyne.

The study is well done and it will provide a new, practical method for the synthesis of oxaboroles. The proposed mechanism is very interesting, but I have some questions

related to the mechanism. I recommend this manuscript for publication provided that the following issues are addressed.

1. The study disclosed that simple alkynes did not react with NaHMDS-HBpin, while propargyl alcohols were reactive. The reason for this result should be discussed.

Author's reply: We appreciate your insightful suggestion. Due to the negative inductive effect of the hydroxyl, the triple bond in propargyl alcohols is more electrophilic than in simple alkynes, making it more reactive toward the nucleophile [HMDS-HBpin]. In contrast, when an electron-donating methoxy group is present on the benzene (Table 2, **6**), the electrophilicity of the triple bond is reduced, resulting in a lower yield. On the other hand, simple alkynes prefer to react with electrophilic BH_3 according to reported base-promoted hydroboration (e.g., ref. 55). We have added this discussion into the substrate scope section as follows: "This highlights the crucial role of the hydroxyl group's negative inductive effect in propargyl alcohols, which increases the electrophilicity of the triple bond, enabling the reaction".

2. In the DFT calculations, the sodium ion occasionally was occasionally included in the intermediate/transition state. In the control experiment, the authors concluded that that the influence of sodium ion is very small. Please explain.

Author's reply: Thank you very much for your suggestion. In our DFT calculations, the inclusion of the sodium ion was indeed necessary to obtain physically reasonable energy barriers. Omitting the sodium ion often led to unrealistic or overestimated activation energies. We apologize if this was not clearly conveyed in the original 3D structural illustrations—this may have caused some confusion. In the revised manuscript and Supplementary Information, we have updated the graphical representations to more clearly and consistently display the presence and coordination of the sodium ion. Please check Fig.6, Figure S3 and Figure S4 again.

Experimentally, the sodium salt is critically important (see also the response to reviewer 2). In contrast, when LiHMDS was added to HBpin in a 1:1 ratio, only a trace of HMDS-Bpin was detected in the ^{11}B -NMR spectrum, indicating that LiHMDS is ineffective in activating HBpin under standard conditions as compared to NaHMDS (see Figure S2 in the Supplementary Information). Moreover, the addition of KHMDS to HBpin in a 1:1 ratio resulted in the formation of a precipitate, showing that the intermediates with potassium counterions are not well soluble; this heterogenous system may hinder the reaction, finally leading to a low yield. This part has been added into the section of mechanistic investigations. In the control experiment, we added 15-crown-5 ether to exclude any specific reactivity of the sodium ion. It means that the sodium ion functions solely as a counter-cation in intermediates and transition states, without exhibiting any additional catalytic activity itself.

Reviewer #4 (Remarks to the Author):

This work realizes a synthesis of five-membered boracycles with moderate to good yields through the anti-addition of pinacolborane to alkynes, catalyzed by the HMDS counteranion.

The manuscript demonstrates that [HMDS-HBpin]- is a key reducing agent in the anti-addition process, as confirmed by various control experiments. Additionally, the authors explore the mechanism behind the preference for the anti-addition pathway and provide an energy profile of the proposed reaction route using DFT calculations. The findings indicate that the steric hindrance from the TMS group results in a higher free energy barrier for the cis-addition process compared to the anti-addition process, explaining the observed preference for the latter when pinacolborane reacts with alkynes. From an application standpoint, the ability to conduct the reaction on a gram scale and the variety of downstream transformations highlight the potential utility of this boracyclic framework construction strategy. Overall, this work is innovative, and I believe the manuscript could be appropriate for publication in Nat. Commun. after addressing the following points:

1. In Fig. 4, the ratio of NaHMDS to HBpin appears inconsistent with the experimental data in the SI. In the section titled "Representative procedure for the NMR titration experiment, Fig. 4a-c" the authors do not specify the concentration of HBpin, making it unclear how to determine the NaHMDS/HBpin ratio in Fig. 4a-c. Furthermore, in the "Representative procedure for Fig. 4e" a mixture of NaHMDS (2 M in THF, 0.2 mmol, 100 μ L) and HBpin (0.2 mmol, 30 μ L) was used, resulting in a NaHMDS/HBpin ratio of 1:1, rather than the 1:2 indicated in Fig. 4e.

Author's reply: We appreciate this insightful suggestion. The HBpin was used in its pure form (neat, undiluted) throughout the experiments. To address this concern, we have added the molar amount of HBpin in parentheses (see SI, Fig. 4a-c). In the first step, we added 0.2 mmol HBpin. Subsequently, 0.08 mmol HBpin was supplied. This resulted in a NaHMDS/HBpin ratio of 1:2.8. Finally, an extra 0.02 mmol of HBpin was introduced to achieve a final NaHMDS/HBpin ratio of 1:3. In the "Representative procedure for Fig. 4e", the amount of NaHMDS should be 0.1 mmol (50 μ L). We have corrected this mistake and apologize for the oversight. Thanks again for your meticulous attention to this detail.

Response to the reviewers

Reviewer #2 (Remarks to the Author):

This reviewer agrees that the authors properly addressed comments raised by all reviewers. The revised mechanism discussion would fully convince readers. I recommend that the authors incorporate part of their response about "bioisostere" to Comment 1 from Reviewer #2 into the revised manuscript. After this very minor revision, I fully agree with publication of this manuscript in Nat. Commun.

Our reply: Thank you very much for your suggestion and support. We have incorporated the part into the first paragraph of the revised manuscript. Please check the yellow part.

Reviewer #3 (Remarks to the Author):

The authors responded and modified the manuscript based on the comments of the reviewers appropriately, and now I recommend this manuscript for publication.

Our reply: Thank you very much for your useful advice and recommendation.

Reviewer #4 (Remarks to the Author):

I have thoroughly examined the submission of this manuscript and am pleased to report that the authors have diligently revised it, I am now very satisfied and the current format will be acceptable to me.

Our reply: Thank you again for your kind suggestions and support.